# Green Extraction Method: Microwave-Assisted Water Extraction Followed by HILIC-HRMS Analysis to Quantify Hydrophilic Compounds in Plants

**DOI:** 10.3390/metabo15040223

**Published:** 2025-03-25

**Authors:** Alexandra Louis, Jean François Chich, Hadrien Chepca, Isabelle Schmitz, Philippe Hugueney, Alessandra Maia-Grondard

**Affiliations:** 1SVQV, INRAE—Université de Strasbourg, 68000 Colmar, France; alexandra.louis2@gmail.com (A.L.); jean-francois.chich@inrae.fr (J.F.C.); hadrien.chepca@inrae.fr (H.C.); philippe.hugueney@inrae.fr (P.H.); 2Université Rouen Normandie, INSA, CNRS, BS UMR 6270, Polymers, Biopolymers, Surfaces Lab., 76000 Rouen, France; isabelle.schmitz@cnrs.fr; 3Université Rouen Normandie, INSERM US 51, CNRS UAR 2026, HeRacLeS PISSARO, 76000 Rouen, France

**Keywords:** primary metabolism, amino acids, organic acids, hydrophilic interaction liquid chromatography (HILIC), microwave-assisted extraction (MAE), high-resolution mass spectrometry

## Abstract

**Background:** Hydrophilic compounds, such as amino acids, organic acids and sugars, among others, are present in large amounts in plant cells. The analysis and quantification of these major hydrophilic compounds are particularly relevant in plant science because they have a considerable impact on the quality of plant-derived products and on plant–pathogen relationships. Our objective was to develop and validate a complete analysis workflow combining a water-based extraction procedure with a fast separation using hydrophilic interaction liquid chromatography coupled to high-resolution mass spectrometry (HILIC-HRMS) for quantitative analysis of hydrophilic compounds in plant tissues. **Results:** Water-based microwave-assisted extraction (MAE) methods for hydrophilic compounds were compared using HILIC-HRMS. The newly developed method involved 20 s MAE time followed by a 10 min HILIC-HRMS analysis. This bioanalytical method was validated for 24 polar metabolites, including amino acids, organic acids, and sugars, to ensure the reliability of analytical results: selectivity, limits of detection and quantification, calibration range and precision. Depending on the compounds, quantification limit was as low as 0.10 µM up to 4.50 µM. Between-run RSDs evaluated on *Vitis vinifera* and *Arabidopsis* samples were all below 20% except for three compounds. **Conclusions:** A water-based MAE method, coupled with HILIC-HRMS, was developed for the absolute quantification of free amino acids, organic acids, and sugars in plant tissues. Its effectiveness was demonstrated in both lignified plants, such as *Vitis vinifera*, and non-lignified plants, such as *Arabidopsis*. This method is suitable for medium- to high-throughput analysis of key polar metabolites from small amounts of plant material.

## 1. Introduction

Plant metabolism can be broadly divided into primary metabolism directly involved in growth and secondary metabolism, also called specialized metabolism, which plays a key role in the plant’s interaction with its environment. The primary metabolism pathways synthesize essential molecules for normal physiological growth and energy requirements of plants. Since 90% of the weight of plant tissues is constituted of water, many of the metabolites produced by these pathways are hydrophilic. Hence, hydrophilic molecules such as amino acids, organic acids and sugars, among others, are an important part of the total compounds that can be extracted from plants. Analysis and quantification of these major polar metabolites is particularly relevant in plant sciences since they greatly impact the quality of plant-derived products. Moreover, free amino acids, organic acids and sugars play a central role in plant–pathogen interactions, as they are used by pathogens for their own metabolism. Therefore, quantification methods allowing an efficient monitoring of major polar metabolites in plant tissues are particularly relevant to study many aspects of plant development, as well as the impact of pathogens on plant health [1,2,3].

Numerous methods have been developed for the quantification of polar molecules using gas chromatography–mass spectrometry (GC-MS) [4,5,6], capillary electrophoresis–mass spectrometry (CE-MS) [7,8] or liquid chromatography–mass spectrometry (LC-MS) [9,10,11]. The chemical derivatization of polar molecules is necessary prior to analysis by GC-MS and may also be used for analysis by LC-MS [12,13,14,15]. Although GC-MS-based methods show high sensitivity and excellent resolution, their main drawback is the need of laborious derivatization procedures. One of the advantages of CE-MS and LC-MS is that polar compounds can be analyzed without derivatization. Short analysis times, selectivity and little matrix interference have been reported for the CE-MS method. However, CE combined with electrospray ionization (ESI)-MS is not easy, due to technical issues and conflicting separation and detection conditions. Only a limited number of methodologies for the online implementation of CE-MS for biological analyses have so far been developed. Two main approaches are commonly used for metabolic profiling using LC-MS: reversed-phase liquid chromatography (RPLC) and hydrophilic interaction liquid chromatography (HILIC). RPLC is the most common technique used to analyze non-volatile components in biological matrices. However, low molecular weight polar compounds such as hydrophilic amino acids or organic acids are usually not sufficiently retained on RPLC columns, resulting in poor chromatographic resolution. Therefore, HILIC constitutes a valuable alternative to RPLC in the field of metabolomics, due to its ability to retain a greater portion of the polar metabolome [15,16,17].

Microwave-assisted extraction (MAE) methods are increasingly more popular as an alternative extraction method in medical and plant studies [18,19]. Microwaves are a form of non-ionizing electromagnetic energy in the 300 MHz to 300 GHz frequency range. This energy is transmitted as waves that can penetrate biomaterials and excite polar molecules. This process has numerous interests owing to its heating mechanism, its moderate cost and good performance under standard atmospheric conditions [20,21]. Moreover, MAE reduces both extraction time and solvent consumption. It is therefore considered as a green extraction method, since low amounts of solvents are needed. For example, El-Malah et al. [22] compared ultrasound and microwave extraction of tomato in order to find the best extraction yield while preserving antioxidant activity. They concluded that MAE was an optimum method to extract lycopene in a low-cost and eco-friendly way. To combine the right analytical conditions with a green extraction procedure, we developed an MAE method based on the results of Carrera and al. [23]. These authors extracted amino acids (AA) from grapes with an ultrasound-assisted procedure. They concluded that water is as effective as, or even better than, methanol as an extraction solvent for most AAs. Thus, the development of water-based MAE could be a safe and inexpensive approach that can achieve high extraction rates with good reproducibility.

To meet the need for optimized HILIC-HRMS applications, we present a new procedure for the extraction, separation and quantification of hydrophilic compounds from plant tissues. This method offers high specificity and sensitivity while being time-efficient. The water-based MAE assay was applied to extract polar metabolome from *Arabidopsis thaliana* leaves, enabling the identification of changes in the composition of polar organic compounds at different stages of rosette development. This environmentally friendly extraction method, combined with a robust analytical procedure, is highly suitable for medium- and high-throughput analyses of the polar metabolome across various plant tissues. Absolute quantification provides precise metabolite concentrations, offering deeper biological insights into metabolic flux analysis, pathway regulation and cross-species comparisons. This makes it a valuable tool for applications in plant breeding, biotechnology, and agronomy. Therefore, this approach holds significant potential for plant development studies and for exploring plant responses to biotic or abiotic stresses [19,24].

## 2. Materials and Methods

### 2.1. Reagents

Amino acid standards were obtained from Sigma–Aldrich (St. Louis, MO, USA) and Wako Chemicals USA, Inc. (Richmond, VA, USA). HPLC-grade acetonitrile was obtained from Honeywell Burdick & Jackson (Morristown, NJ, USA). All other reagents and standards were obtained from Sigma–Aldrich.

### 2.2. Leaf Samples

*Vitis vinifera* (grapevine), *Nicotiana benthamiana* (tobacco) and *Arabidopsis thaliana* leaves from the collection of the INRAE Colmar were frozen at −20 °C until analysis. For analysis of senescence in *A. thaliana*, the plants were grown in a growth chamber under 20 ± 1 °C and 14 h photoperiod under fluorescent lamps during 3 weeks. Leaves from the different levels were detached and immediately frozen in liquid nitrogen. For the microscopy analysis, we used fresh leaves from the collection of the INRAE Colmar.

### 2.3. Microcopy Analysis

Observations were made using a microscope (Zeiss AxioZoom.V16) with a PlanNeoFluar Z 1x/0.25 FWD 56 mm objective lens. Samples were illuminated by a Zeiss CL4500 LED device. Images were analyzed using ImageJ (Version 2.14.0/1.54f) [25] and figures were mounted using the plugin FigureJ (Version 1.10b) [26]. Bars represent 100 µm.

### 2.4. Preparation of Leaf Samples

The grapevine and *Arabidopsis* leaf samples were frozen, lyophilized and homogenized using a bead mill. Two milligrams of grapevine leaf powder was used to determine the best combination of irradiation and time for MAE. Thus, 2 mg was submitted to MAE in water (400 µL) for different times (15 s, 20 s, 30 s) and irradiation powers (300 W, 450 W, 600 W). For application of the selected method, 2 mg *Arabidopsis* leaf powder was extracted with 400 µL of water in an Eppendorf tube, and irradiation was performed at 450 W and 20 s. Tubes were centrifuged and supernatant was analyzed using high-performance liquid chromatography–high-resolution mass spectrometry with electrospray ionization (UHPLC-ESI-HRMS) in the conditions described below.

### 2.5. Optimizing Microwave-Assisted Extraction (MAE)

Polar compounds from grapevine leaves were extracted using a domestic microwave oven system (Samsung MW 120 N). The apparatus was equipped with a control system for irradiation time and microwave power (the latter was adjustable by discrete increments from 100 to 1000 W). We optimized the extraction protocol using two different and successive assays: (1) extraction volume first and (2) time and irradiation conditions. For the first step, two milligrams of the leaf sample was placed into a 1.5 mL Eppendorf tube. Different water volumes (100, 200, 300, 400 µL) were added to the leaves and the tubes were placed in the microwave-irradiation zone with fixed irradiation and time, 600 W and 15 s, respectively. For the second step, two milligrams of leaf samples and the optimized volume of water were submitted for different powers (300, 450 or 600 W) and times (15, 20 or 30 s) to microwave irradiation. For each test, three replicates were performed. After vortexing a few seconds, the extract was centrifuged at 12,000× *g* for 3 min to minimize interphase thickness and to separate aqueous extract form leaf particles. The supernatant was recovered and analysed using HILIC-HRMS.

### 2.6. HILIC-HRMS

The analysis of water extracts was performed using an UHPLC system (Dionex Ultimate 3000; Thermo Fisher Scientific, San Jose, CA, USA) equipped with a diode array detector (DAD). The chromatographic separation was performed on a Nucleodur HILIC column (100 mm length, 2 mm i.d., 1.8 µm particle size; Macherey-Nagel, Germany) at 30 °C. The mobile phase consisted of acetonitrile/formic acid (0.1%, *v*/*v*) (eluent A) and water/formic acid (0.1%, *v*/*v*) (eluent B) at a flow rate of 0.30 mL min^−1^. The gradient elution program was as follows: 0–1.2 min 2% B; 1.2–4.2 min 2–50% B; 4.2–7.2 min 50% B; 2.5 min of the column restoration. The sample volume injected was 1 µL. The liquid chromatography system was coupled to an Exactive Orbitrap mass spectrometer (Thermo Fisher Scientific) equipped with an electrospray ionization source operated in positive or negative ionization mode. Parameters were 400 °C for ion transfer capillary temperature and 2500 and 3400 V for negative and positive needle voltages, respectively. Nebulization with nitrogen sheath gas and auxiliary gas were maintained at 40 and 5 arbitrary units, respectively. The spectra were acquired within the *m*/*z* mass range of 75–1500 atomic mass units (a.m.u.), using a resolution of 50000 at *m*/*z* 200 a.m.u. The system was first externally calibrated using the standard mixture from Thermo Fisher Scientific. In positive ion mode, the system was internally calibrated using dibutyl phthalate as a lock mass (*m*/*z* 279.1591) giving a mass accuracy below 1 ppm. Peak integration was performed with the Xcalibur software (version 4.7).

### 2.7. Statistical Analysis

For all measurements, three leaf replicates were extracted and analyzed independently per each stage of development and validation of the methodology. Statistical analysis was performed using Tukey’s honest significant difference method followed by a false discovery rate (FDR) correction, with FDR < 0.05, using the software R version 3.5.1 [27] for the application in different pairwise comparison of the biological matrices. For analysis of the combination between power irradiation and times using MAE, the optimal combination was determined and validated using response surface methodology using the software R version 3.5.1 [27].

### 2.8. Validation of the Method

After defining the best condition for MS analysis and sample dilutions, the HILIC-HRMS method was validated in terms of selectivity, precision, linearity and limits of detection (LOD) and quantification (LOQ).

### 2.9. Linearity for Amino Acids, Organic Acids and Sugars Analysis

Amino acids, organic acids and sugars quantifications were based on calibration curves obtained with the respective standard. The standard solutions of amino acids, organic acids and sugars were prepared at concentrations on the order of µM for calibrations (Table 1). Results were used to calculate overall linearity together with precision at each concentration.

### 2.10. Selectivity, Sensitivity and Reproducibility of the Method

The selectivity is evaluated along with the retention time and the precise mass measurement for each analyte. The limits of detection (LOD) and quantification (LOQ) were assessed by determining the lowest starting concentration that resulted in a signal-to-noise ratio of ≥3 (LOD) and ≥10 (LOQ) (Table 1). Method reproducibility was evaluated by preparing and analyzing three independent biological replicates of water-extracted samples from two different leaves. Each sample was analyzed three times (Appendix A).

## 3. Results

### 3.1. Optimizing Hydrophilic Metabolite Extraction with MAE

The initial assays were performed with grapevine leaves, because these tissues were likely to be quite resistant to extraction procedures, due to the ligneous nature of this plant. The water volume appropriate for the extraction of 2 mg leaf discs (dry weight) was determined based on published literature and preliminary experiments. Different reports on the relationship between tissue mass and extraction volume have been published [18,28,29,30]. Based on these indications, the objective was to apply an extraction volume that met the dual challenge of optimizing the detection of metabolites with different chemical properties while remaining within the dynamic measurement range of the UHPLC-HRMS instrument (Table 1). Following preliminary experiments, a volume of 400 µL of water was selected for the optimization of the MAE procedure. This relatively high extraction volume allowed for easy pipetting and handling of samples, enhancing the robustness of the method (see Appendix A). Finally, as shown in the validation section, the LOD and LOQ values were sufficiently high to address our research questions.

The design of an experimental plan to determine the best combination of irradiation and time for MAE was conducted with the homogenized grapevine leaf discs. Thus, 2 mg was submitted to MAE in water (400 µL) for different times (15 s, 20 s, 30 s) and irradiation powers (300 W, 450 W, 600 W). The optimal combination was determined using response surface methodology (RSM). The results show a maximum efficiency for an extraction at 450 W for 20 s, this efficiency being represented by the sum of the responses of the compounds to each modality (Figure 1).

The effect of the extraction procedures on leaf structure was observed by optical microscopy (40× magnification). Grapevine leaf discs of 1 cm diameter (~2 mg) were submitted to MAE and UAE under the conditions described above. Microscopy observations did not show major structural modification of grapevine leaves (see Appendix A).

### 3.2. Validation of an HILIC-Based Method for the Separation and Quantification of Major Polar Metabolites from Plants

The main characteristics of a bioanalytical method that are essential to ensure the acceptability of the performance and the reliability of analytical results are selectivity, limits of detection and quantification, a linear dose/response and good calibration range (calibration curve performance), and precision of the analyses. The chromatographic conditions of the HILIC column used in this work were based on Xiaolong et al. [16]. These conditions gave good chromatographic performance for the HRMS-based quantification of the selected compounds of interest in various plant tissues, including grapevine and *Arabidopsis* leaves. Nevertheless, chromatographic conditions may be optimized for specific applications in future work.

#### 3.2.1. Selectivity

The first step of the method validation consisted in the evaluation of the performance of an HILIC column operated at 30 °C with a gradient of water (0.1% acid formic) and acetonitrile (0.1% acid formic) as a mobile phase. The selected chromatographic conditions allowed fast separations, with run times less than 10 min, including return to initial conditions. Commercially available underivatized standards were used to determine retention times for a selection of major polar metabolites usually found in plant tissues, including organic acids, amino acids and sugars (Table 1). Using these chromatographic conditions, retention times were set between 2 and 6 min, allowing good separation and quantification of the selected standards, owing to accurate mass measurements (mass accuracy better than 2 ppm in negative ion mode and better than 1 ppm in positive ion mode). In addition, the separation efficiency of the column allowed for separating fructose and glucose. Therefore, selectivity was ensured owing to retention time and mass measure accuracy for each metabolite quantified. Blank samples were injected to confirm that no carry-over would occur from one injection to another. Indeed, no carry-over was observed in any of the assays.

#### 3.2.2. Limits of Detection and Quantification, and Linearity

The limits of detection (LOD) and quantification (LOQ) of each compound were determined in ESI^+^ and ESI^−^ analyses on the basis of a signal-to-noise (S/N) ratio better than 3 and 10, respectively. The results for the optimal analysis conditions are reported in Table 1. The LOQ ranged from 0.10 µM for sucrose, serine and asparagine, to 4.50 µM for malic acid. The dose–response relationships of the selected standards were established in ESI^+^ and ESI^−^ analyses. The working range was up to 512.7 µM for tartaric acid. Linear regressions were performed for each of these compounds, resulting in regression coefficients R^2^ from 0.989 to 0.999 over the indicated concentration ranges, with most values above 0.995.

#### 3.2.3. Precision

Lastly, the within-run precision of the LCMS method was assessed. The relative standard deviation (RSD) of each amino acid, organic acid and sugar was measured by three consecutive 1 µL injections of the standard mixture at 10 µM. The RSDs were lower than 8% for all compounds, the average RSD being 3.6% ± 1.7% for amino acids, 7.4% ± 1.9% for organic acids and 5.1% ± 0.9% for sugars (Table 1). For the analysis of the relative standard deviation (RSD) of polar metabolites in the application of the MAE method to leaf tissues of *Vitis vinifera* and *Arabidopsis*, inter-series precision was evaluated through injections of three biological replicates. The RSDs were calculated based on the peak area measurements of each compound, resulting in RSDs below 20% for the majority of compounds, except for threonine, tryptophan and valine. Indeed, for these compounds, the RSDs ranged between 20% and 24% (Appendix A).

### 3.3. Water-Based MAE Combined with HILIC Chromatography Applied to the Analysis of the Polar Metabolome in Rosette Leaves of Arabidopsis at Distinct Developmental Stages

Leaf development and senescence is an essential process for plant physiology and metabolism [29]. The modification of the primary metabolome has been thoroughly studied in *Arabidopsis* rosette leaves at different developmental stages by Watanabe et al. [29]. To validate the relevance of the MAE/HILIC-HRMS analysis method, we analyzed individual leaves of *Arabidopsis* at three different developmental stages (Appendix A). Developmental stages were as follows: stage 1 (N1 = 0.5–1 cm) leaves were 10% expanded, stage 2 (N2 = 2–3 cm) leaves were 50% expanded and stage 3 (N3 = 4–5 cm) were fully expanded.

All of the compounds previously described were detected in each matrix, showing the general applicability of the method for accurate and fast extraction of hydrophilic compounds from leaves at various development stages. To obtain a global view, principal component analysis (PCA) was performed (Figure 2A). Out of 24 metabolites searched, 22 were detected, amino acids, organic acids and sugars. The first principal component (PC1), accounting for 57.2% of the total variance, resolved the time series of leaf development. In particular, the metabolites of the first stage N1 were resolved from stage N3 in PC1 and from stage N2 in PC3, accounting for 9% of the total variance, resolving the physiological phase changes. Thus, PCA can discriminate between various stages and displays sequential changes that indicate coordinated physiological processes (Figure 2B).

Differential analysis was then carried out to identify differences between stages on the selected metabolites. By comparing N2 and N3 to the youngest leaf N1, significant differences in the relative amounts of these metabolites were observed (Appendix A).

## 4. Discussion

Analysis and quantification of major polar metabolites is of high interest to study many aspects of plant physiology and development, as well as interaction with their biotic and abiotic environment. Efficient monitoring of polar metabolites in plant tissues requires optimized extraction procedures combined with robust analytical methods. The traditional methanol-based UAE is often used for the extraction of polar compounds, but it has a low selectivity for highly polar metabolites. Due to this low selectivity and contamination by other non-hydrophilic compounds, retention time reproducibility is poor when this extraction method is used in combination with HILIC chromatography. In addition, problems of increased column back-pressure occur after large series of samples [18], resulting in a degradation of the analytical performances. Therefore, in this work, we decided to develop a green, water-based extraction method using MAE. The aim of this work was to develop an optimized water-based extraction procedure combined with a fast and reliable HILIC-HRMS analytical method, suitable for the analysis of major polar metabolites in plants.

### 4.1. UAE and MAE Mechanisms Comparison

The mechanisms in UAE and MAE are very different. In UAE, the application of sonic energy to agitate particles of a sample allows the extraction of multiple compounds from plants. However, it has many chemical and physical drawbacks. In MAE, non-ionizing electromagnetic energy occurs at frequencies ranging from 300 MHz to 300 GHz. This energy is transmitted in the form of waves, which can penetrate biomaterials and interact with polar molecules of the materials, such as water, to generate heat spots [23].

Microwave energy acts directly on molecules through ionic conduction and dipole rotation. Thus, only polar materials can be heated, depending on their dielectric constant. The electric field of the microwave radiation interacts with the material and causes heating, affecting molecular parameters, such as dipole rotations or ion migration [23]. However, they cannot ionize atoms or change molecular structures due to their low energy content.

The assumptions that “the weight of the leaves is largely water and therefore the leaf blade is mostly water” [31] still seem valid. Indeed, only a few studies have addressed differences in leaf water content between different plants or their changes under varying environmental conditions. Therefore, microwave extraction is performed from the inside out. This feature allows the extraction of active compounds from the plant by internal overpressure and osmotic effect [18]. The result is a rapid, highly efficient extraction without organic solvents, with highly concentrated metabolites. Microscopic observation of leaf tissues subjected to the MAE or UAE methods did not reveal any significant changes in the ultrastructure of grapevine leaves, regardless of the time or microwave power (Appendix A). Ultrasound seems more aggressive since the appearance of the leaves was different from the control and the microwave-treated leaves. To visualize this, we tested with less lignified plants such as *Arabidopsis* and *N. benthamiana*. They show similar sensitivity to microwave treatment. At 350 W for 15 s, both plants keep an ultrastructure similar to the control. When the power and irradiation time are increased, the structures are affected. The cell shapes become irregular and the tobacco is bleached. Ultrasound treatment affects *N. benthamiana* but not *Arabidopsis*.

The differences in leaf sensitivity to microwave treatment are likely related to whether the plant is herbaceous or woody. The structure of grape leaves, being from a woody plant, appears more resistant to physical treatments compared to the leaves of herbaceous plants such as *Arabidopsis* and *N. benthamiana*. Nevertheless, when comparing these observations with extraction measurements, it can be concluded that extraction efficiency cannot be inferred solely from microscopic phenotypic observations.

### 4.2. Combination of Water-Based MAE with HILIC-HRMS for Efficient Characterization of Polar Metabolites in Plant Tissues

The chromatographic condition of HILIC columns was developed based on Xiaolong et al. [16], giving good chromatographic performance for HRMS-based quantification of the selected compounds of interest, while allowing fast separations times under 10 min. Water molecules are attracted to the polar groups of different types of stationary phases, forming an aqueous layer on the surface. Consequently, a polar analyte dissolved in the mobile phase partitions between this semi-immobilized aqueous layer and the mobile phase, which contains some aqueous content. The more hydrophilic the analyte, the more the partitioning equilibrium shifts toward the immobilized water layer on the stationary phase, leading to greater retention of the analyte. In addition to partitioning, hydrogen-donor interactions and weak electrostatic mechanisms have also been observed, indicating that both adsorption and partitioning play important roles in retention in the HILIC mode. This is due to intermolecular forces, such as electrostatic interactions, hydrogen bonding, dipole–dipole interactions and weak hydrophobic interactions, which have been highlighted in numerous studies on HILIC separation.

The parameters of ESI ionization were validated using several underivatized standards. The selected chromatographic conditions allowed fast separation with a run time less than 10 min, including return to initial conditions. The duration of the method is competitive with recent published studies [14,16,32] to answer to the issue of the implementation of high-throughput analyses. The performances of our optimized method are reported in Table 1. Linear relationships were observed between peak areas of extracted ion chromatograms (EIC) of precise *m*/*z* values and concentrations for all the studied compounds. The concentration ranges are adapted to the quantification of these metabolites in plant tissues (Table 1). These ranges are notably relevant to the biological context, where metabolite amounts are often in the range of micromolar to millimolar concentrations. Indeed, the measured LOQ presented the greatest interest in the high-resolution measurements for improving the sensibility of the method, with most values lower than the LOQ described with the traditional multiple reaction monitoring mode [32], after a similar HILIC separation method. In comparison to the sensitivity of the derivatization procedure, sensitivity remained slightly better after derivatization [14], but the method described here does not involve potentially harmful derivatization agents. No carry-over was observed in any of the tested samples, indicating that this method is appropriate for analyzing large numbers of samples involving complex biological matrices. Lastly, the within-run precision of the LCMS method was assessed. The relative standard deviation (RSD) of each amino acid, organic acid and sugar was measured by three consecutive 1 µL injections of the standard mixture at 10 µM. The RSDs were lower than 8% for all compounds: the average RSD being 3.6% ± 1.7% for amino acids, 7.4% ± 1.9% for organic acids and 5.1% ± 0.9% for sugars (Table 1).

For the optimization of hydrophilic metabolite extraction with MAE, a small amount of homogenized leaf material (2 mg dry weight leaf discs) was prioritized, as this best corresponds to the biological experiments conducted in our laboratory. The design of an experimental plan to determine the best combination of irradiation and time for MAE was based on the literature and our particular constraints. These constraints include the use of a microwave with a power range between 100 and 600 W, together with the need to identify an application time that is both efficient and rapid for processing a large volume of samples. For example, this design is applied to phenotypic profiling in mQTL research, where hundreds of samples are analyzed.

Based on this context, the analyses for the plan were conducted using homogenized grapevine leaf discs (2 mg). These samples were subjected to MAE in water (400 µL) for varying durations (15 s, 20 s, 30 s) and irradiation powers (300 W, 450 W, 600 W). The optimal combination was determined using RSM, a statistical approach designed to identify optimal operating conditions through experimental techniques. In this method, variables are correlated through polynomial functions based on the Stone–Weierstrass theorem [33]. The RSM results indicated that the optimal extraction yield was achieved with 450 W for 20 s (Figure 2). Indeed, as indicated in the validation section, the LOD and LOQ values were high enough to meet our needs. In addition, this relatively large extraction volume increases the robustness of our method.

Using these MAE conditions, the response was stable and matrix effects were minimized, thus increasing the robustness of the method. For the analysis of the relative standard deviation (RSD) of the selected hydrophilic metabolites in the application of the MAE method to leaf tissues of *Vitis vinifera* and *Arabidopsis*, inter-series precision was evaluated through injections of three biological replicates. The RSDs were calculated based on the peak area measurements of each compound, resulting in RSDs below 20% for most of the compounds (Appendix A), except for threonine 20%, valine 23% in *Arabidopsis* and tryptophan 24% in *Vitis vinifera*.

### 4.3. Application of the MAE /HILIC-HRMS Method to the Analysis of the Polar Metabolome of Arabidopsis Rosette Leaves

Amino acids are major compounds involved in all aspects of plant physiology [30,34,35]. As a consequence, amino acid metabolism and catabolism involve a wide set of general and specific regulators and show significant differences among plant species, tissues and developmental stage [35]. Rosette leaves of *Arabidopsis* constitute a well-described system for monitoring development-related changes in primary metabolites [29,30]. To validate our MAE /HILIC-HRMS analytical method, we analyzed the polar metabolome of *Arabidopsis* rosette leaves at different developmental stages: young (N1), intermediate (N2) and fully developed (N3) (Appendix A). To obtain a global outlook, principal component analysis (PCA) was performed with the 22 metabolites quantified, which include amino acids, organic acids and sugars (Figure 2A). The first dimension (Dim 1), accounting for 57.2% of the total variance, resolved the time series of leaf development. Thus, PCA corroborates major changes among developmental stages and displays sequential changes indicating coordinated processes.

Analysis of metabolite contribution for the discrimination N1 and N3 in Dim 1 (Figure 2B) highlighted the strong contribution of arginine and methionine. Arginine is a precursor for the synthesis of polyamines (putrescine, spermidine, and spermine), which are essential for the development and stress responses of plants [36]. The sulfur-containing amino acid methionine is essential in all organisms as a building block of proteins and as a component of the universal activated methyl donor S-adenosylmethionine [37].

Among the twelve metabolites with the most important contribution to the PCA (Figure 2B) were isoleucine, valine and leucine for the branched chain amino acids (BCAAs); tryptophane, tyrosine and phenylalanine for the aromatic amino acids (AAAs); and asparagine, aspartate, glutamine and glutamate for another group of amino acids involved in carbon and nitrogen metabolism.

We observed a decrease in the contents of the BCAAs and AAAs (Figure 3B,C). The results indicated good plant growth conditions with no BCAAs in N3, as an alternative respiration support, in the case of dark-induced senescence [38,39]. We did not detect any accumulation of AAAs, which are an important source of precursors to support the synthesis of the large number of specialized metabolites, including stress-related flavonoids [40,41].

Another group of AAs (Figure 3A) are major long-distance transport forms of both carbon and nitrogen in xylem and phloem [42]. Asparagine (Asn) and glutamine (Gln) did not show significant difference in N1, N2 and N3, unlike aspartate (Asp) and glutamate (Glu), with decreased amounts along leaf development compared with N1. This resulted in an increase in Gln/Glu ratio and without modification of the Asn/Asp ratio. Their conversion products have long been documented to be crucial for nitrogen transport [29].

A differential analysis was then carried out to monitor the impact of leaf development on the selected metabolites. Comparison of N2 and N3 to the youngest leaf N1 highlighted a general decrease in amino acid and organic acid contents during leaf development, with the noticeable exception of serine (Appendix A). The results are in good agreement with those obtained by Kanojia et al. [30], indicating that the MAE/HILIC-HRMS analytical method presented here is robust and reproducible, and well suited for monitoring hydrophilic metabolites in plant tissues. RSDs were calculated based on the peak area measurements of each compound, resulting in RSDs below 20% for most of the compounds (Appendix A), except for threonine and valine 20% and 23%.

## 5. Conclusions

Here, we present the development of a new method for the extraction, separation and analysis of polar compounds in plant tissues. This eco-friendly approach is based on microwave-assisted extraction (MAE) in water, followed by HILIC-HRMS for metabolite separation and quantification. The separation step was validated using commercially available standards. The entire method proved to be simple, fast, sensitive, green and reproducible, making it suitable for analyzing small tissue samples. The extraction results from various plant tissues, both herbaceous and woody, demonstrated remarkable reproducibility, indicating that our method is well suited for a wide range of plant materials. As a validation, a differential analysis of hydrophilic metabolites in *Arabidopsis* leaves at various developmental stages successfully reproduced previously published results. Overall, the simplicity, speed, sensitivity and reproducibility of this approach make it a method of choice for studying primary metabolism in plants.

## Figures and Tables

**Figure 1 metabolites-15-00223-f001:**
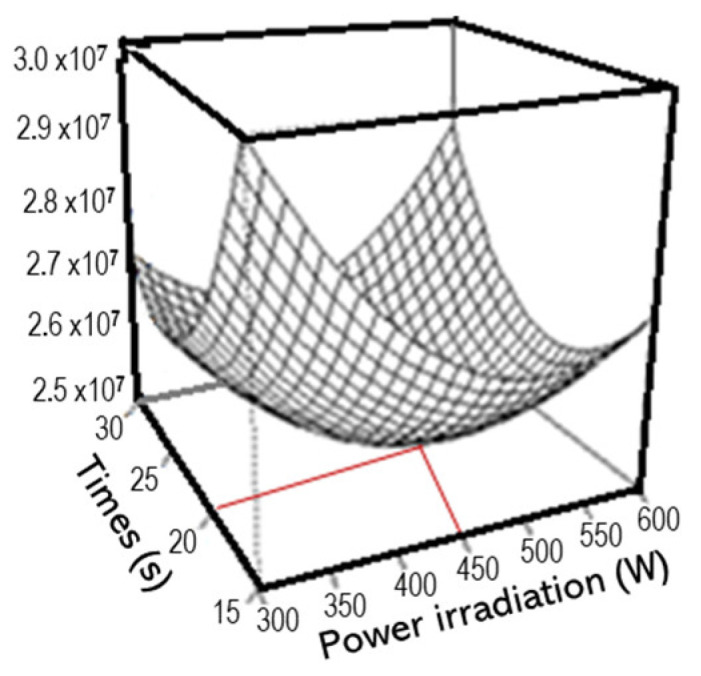
Analysis of the combination between power irradiation and times using MAE. Data represent the addition of metabolites in each three replicates for each combined condition. The red line represents the best condition.

**Figure 2 metabolites-15-00223-f002:**
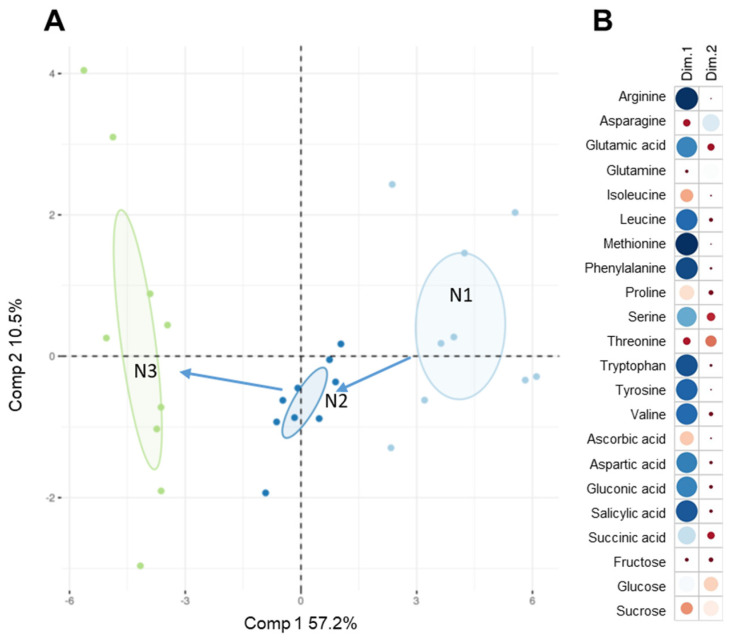
(**A**) Principal component analysis (PCA) was performed on all analyzed compounds in spatiotemporal distinct stages (N1, N2 and N3) of *Arabidopsis* rosette leaves. The plot was constructed with the 22 annotated metabolites, including amino acids, organic acids and sugars. Data represent seven to nine biological replicates for each point. Components 1 and 2 explain 57.2% and 10.5% of the variance, respectively. (**B**) Contribution of each metabolite to PCA components is relative to the circle size and the color intensity, the larger and darker ones contribute more.

**Figure 3 metabolites-15-00223-f003:**
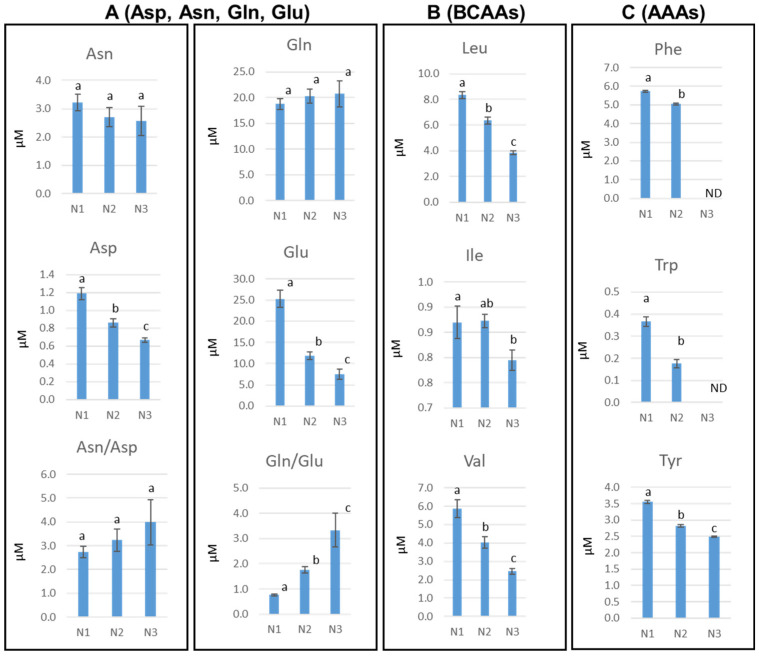
Changes in the selected metabolites in spatiotemporal distinct stages (N1, N2, N3) in *Arabidopsis* rosette leaves after performing MAE in water extraction. Data represent mean values of seven to nine biological replicates for each condition and time point. Error bars represent SD. Different letters represent statistically significant differences (*p* < 0.05) using Tukey’s test. Abbreviation of metabolites: asparagine (Asn), glutamine (Gln), aspartate (Asp) and glutamate (Glu).

**Table 1 metabolites-15-00223-t001:** Response characteristics of the AA, OA and sugar standards with UHPLC-ESI-HRMS using an HILIC column. (* ESI^+^ [M+H]^+^, others in ESI^−^ [M-H]^−^). For each compound, detected *m*/*z* and accuracy (in ppm) are indicated.

Compounds	RT (min)	(*m*/*z*)	Accuracy (ppm)	Regression Data				
				Linear Range (µM)	R2 Value	LOQ (µM)	LOD (µM)	RSD (%)
Arginine	4.49	173.1035	0.238	2.9–338.0	0.9922	2.90	0.95	3.6
Asparagine	5.09	131.0454	0.281	0.1–445.6	0.9968	0.10	0.04	1.4
Cysteine	5.03	120.0115	1.286	0.8–486.1	0.9978	0.80	0.20	0.9
Glutamine	5.04	145.0608	0.011	0.7–402.9	0.9951	0.70	0.22	2.5
Leucine *	4.24	132.102	0.793	1.5–449.0	0.9972	1.50	0.70	3.7
Lysine	4.56	145.0973	1.418	2.1–402.9	0.9918	2.10	1.80	6.4
Methionine	4.48	148.0429	1.717	1.3–394.8	0.9990	1.30	0.70	3.6
OH-proline	5.15	130.0502	0.32	0.8–56.5	0.9976	0.80	0.25	3.9
Phenylalanine	4.57	164.0709	1.919	1.2–356.5	0.9928	1.20	0.40	2.9
Proline *	4.86	116.0707	1.162	4.3–511.5	0.9912	4.30	2.10	0.3
Serine	5.11	104.0342	0.13	0.1–70.5	0.9929	0.10	0.04	4.2
Threonine *	5.03	118.0498	0.03	0.8–123.5	0.9961	0.80	0.25	3.1
Tryptophan *	4.31	205.0972	0.662	0.5–18	0.9985	0.50	0.05	4.6
Tyrosine	4.49	180.0658	1.946	1.1–325.0	0.9980	1.10	0.50	5.6
Valine	4.42	116.0707	1.076	1.7–251.3	0.9970	1.70	0.80	5.5
Glutamic acid	5.25	146.0448	0.046	0.3–402.9	0.9934	0.30	0.05	4.8
Gluconic acid	5.67	195.0503	2.209	3.05–196	0.9986	3.05	0.90	7.0
Malic acid	5.64	133.0132	0.754	4.5–287.3	0.9893	4.50	2.40	8.1
Salicylic acid	3.23	137.0234	0.945	0.7–69.6	0.9954	0.70	0.04	9.6
Succinic acid	2	117.0184	1.067	2.1–162.7	0.9938	2.10	1.80	7.8
Tartaric acid	5.67	149.0082	1.045	4.0–512.7	0.9972	4.00	1.50	5.1
Fructose	4.53	179.0553	2.097	0.7–92.6	0.9959	0.70	0.03	4.3
Glucose	4.62	179.0553	1.818	0.7–92.6	0.9994	0.70	0.03	6.2
Sucrose	5.03	341.1086	2.44	0.1–97.5	0.9946	0.10	0.03	5.1

## Data Availability

The datasets used and/or analyzed during the current study are available from the corresponding author upon reasonable request.

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
