# Peer review of "Green Extraction Method: Microwave-Assisted Water Extraction Followed by HILIC-HRMS Analysis to Quantify Hydrophilic Compounds in Plants"

_metabolites, 2025, doi:10.3390/metabo15040223_

Round 1
Reviewer 1 Report
Comments and Suggestions for Authors
There several points that need to be addressed;
Abstract
The overall sensitivity and reproducibility for tested compounds was not mentioned. RSD was missing. Comparison should be made to highlight the sensitivity and reproducibility of the method. Results on Arabidopsis rosette and grapevine leaves were not mentioned as well. A summary of the findings should be put in the abstract.
Why use relative quantification? This is an absolute quantification which used pure standard to built the calibration curve for each compound.
Background
What was the reason of doing an absolute quantification?
What was the impact of doing an absolute quantification in this study and namely to study of plant metabolism?
Results
Table 1 was missing RSD. Any intraday injection was carried out?
Results on RSM was included in discussion? Please include the RSM results and how RSM was used to select the best extraction parameter
Method development on HILIC was lacking. How the influence of mobile phase gradient, ionic strength, and column temperature were evaluated?
Validation method using Arabidopsis rosette and grapevine leaves was not well explained. How extraction was carried out? Is water only enough to extract compounds of interest?
Author Response
Comments and Suggestions for Authors
There several points that need to be addressed;
Abstract
The overall sensitivity and reproducibility for tested compounds was not mentioned. RSD was missing. Comparison should be made to highlight the sensitivity and reproducibility of the method. Results on Arabidopsis rosette and grapevine leaves were not mentioned as well. A summary of the findings should be put in the abstract.
Why use relative quantification? This is an absolute quantification which used pure standard to built the calibration curve for each compound.
The sensitivity of the method is mentioned in term of limit of quantification in the abstract. RSD were added based on application results on Vitis vinifera and Arabidopsis. In table 1, the reviewer will find the RSD corresponding to the calibration curve.
This is right. As calibration curves have been obtained with standards for all compounds, the term “absolute quantification” is more adapted. This has been corrected in the abstract.
Background
What was the reason of doing an absolute quantification?
What was the impact of doing an absolute quantification in this study and namely to study of plant metabolism?
To address these questions, lines 95–102 were completed as follows:
This environmentally friendly extraction method, combined with a robust analytical procedure, is highly suitable for medium- and high-throughput analyses of the polar metabolome across various plant tissues. Absolute quantification provides precise metabolite concentrations, offering deep biological insights into metabolic flux analysis, pathway regulation and cross-species comparisons. This makes it a valuable tool for applications in plant breeding, biotechnology, and agronomy. This approach holds therefore significant potential for plant development studies and for exploring plant responses to biotic or abiotic stresses.
Results
Table 1 was missing RSD. Any intraday injection was carried out?
Intraday injections have been carried out. RSD were included in Table 1 and the Results section was modified lines 181-187 as follows:
For the analysis of the relative standard deviation (RSD) of polar metabolites in the application of the MAE method to leaf tissues of Vitis vinifera and Arabidopsis, inter-series precision was evaluated through injections of three biological replicates. The RSDs were calculated based on the peak area measurements of each compound, resulting in RSDs below 20% for the majority of compounds, except for threonine, tryptophan and valine. Indeed, for these compounds, the RSDs ranged between 20% and 24% (Additional file S1).
Results on RSM was included in discussion?
Please include the RSM results and how RSM was used to select the best extraction parameter
The RSM results were included lines 125–128 and lines 324-327 as follows:
The optimal combination was determined using the response surface methodology (RSM). The results show a maximum efficiency for an extraction at 450 W for 20 seconds, this efficiency being represented by the sum of the responses of the compounds to each modality (Figure 2).
The optimal combination was determined using RSM, a statistical approach designed to identify optimal operating conditions through experimental techniques. In this method, variables are correlated through polynomial functions based on the Stone-Weierstrass theorem [47]. The RSM results indicated that the optimal extraction yield was achieved with 450 W for 20 seconds (Figure 2).
Method development on HILIC was lacking. How the influence of mobile phase gradient, ionic strength, and column temperature were evaluated?
The chromatographic conditions of HILIC column used in this work were based on Xiaolong et al. [24]. These conditions gave good chromatographic performance for HRMS-based quantification of the selected compounds of interest in various plant tissues, including grapevine and Arabidopsis leaves. Therefore, the specific influence of the mobile phase gradient, ionic strength, or column temperature on HILIC were not evaluated further in the present work. Nevertheless, such parameters may be optimized for specific applications in future works.
This has been mentioned lines 144 to 149.
Reviewer 2 Report
Comments and Suggestions for Authors
|
In the manuscript of A. Louis et al. “Green Extraction Method: Microwave-Assisted Water Extraction Followed by HILIC-HRMS Analysis to Quantify Hydrophilic Compounds in Plants”, a complete analysis workflow combining a water-based extraction procedure with a fast separation using hydrophilic interaction liquid chromatography coupled to high resolution mass spectrometry (HILIC-HRMS) for quantitative analysis of hydrophilic compounds in plant tissues was developed and validated. Water-based microwave-assisted extraction (MAE) methods for hydrophilic compounds were compared using HILIC-HRMS. The newly-developed method involved 20 s MAE extraction time followed by a 10 min HILIC-HRMS analysis. This bioanalytical method was validated for 24 polar metabolites, including amino acids, organic acids and sugars, to ensure the reliability of analytical results: selectivity, limits of detection and quantification, calibration range and precision. Depending on the compounds, quantification limit was as low as 0.10 microM up to 4.50 microM. The authors believe that this method is amenable for medium to high throughput analysis of major polar metabolites from small amounts of plant material.
Reviewer’s notes.
- Line 115, Table 1. Typos. Should be: ”tartaric acid” “fructose”.
1a. Same, column 3. Are there exact (i.e., calculated) m/z values? If yes, it would be better to recalculate m/z values for these ions (mass of a singly charged ion differs from that of corresponding neurtal in the mass of electron, e.g., for Arg [M–H]– m/z 173.1044, etc.)
- Lines 143 and 144. Should be “formic acid”.
- Line 438. (Tableau … in supplementary data). What is missed?
- Over the text: species names should be italicized.
Author Response
Reviewer’s notes.
- Line 115, Table 1. Typos. Should be: ”tartaric acid” “fructose”.
This has been corrected
1a. Same, column 3. Are there exact (i.e., calculated) m/z values? If yes, it would be better to recalculate m/z values for these ions (mass of a singly charged ion differs from that of corresponding neurtal in the mass of electron, e.g., for Arg [M–H]– m/z 173.1044, etc.)
Column 3 reports the measured m/z, and the accuracy relative to the calculated m/z is noted in column 4 (in ppm). This is now mentioned in the legend of Table 1. The type of ion detected either in positive mode [M+H]+ or in negative mode [M-H]- is now specified in the legend as well.
- Lines 143 and 144. Should be “formic acid”.
- Line 438. (Tableau … in supplementary data). What is missed?
- Over the text: species names should be italicized.
Thank you for your comments, these changes have been made.
Round 2
Reviewer 1 Report
Comments and Suggestions for Authors
All comments have been addressed by the author. I would like to recommend this manuscript for publication.